# Differential Expression of Genes at Panicle Initiation and Grain Filling Stages Implied in Heterosis of Rice Hybrids

**DOI:** 10.3390/ijms21031080

**Published:** 2020-02-06

**Authors:** Jawahar Lal Katara, Ram Lakhan Verma, Madhuchhanda Parida, Umakanta Ngangkham, Kutubuddin Ali Molla, Kalyani Makarand Barbadikar, Mitadru Mukherjee, Parameswaran C, Sanghamitra Samantaray, Nageswara Rao Ravi, Onkar Nath Singh, Trilochan Mohapatra

**Affiliations:** Crop Improvement Division, ICAR-National Rice Research Institute (Formerly CRRI), Cuttack-753006, Odisha, India; jawaharbt@gmail.com (J.L.K.); ram.pantvarsity@gmail.com (R.L.V.); mitanyss@gmail.com (M.P.); ukbiotech@gmail.com (U.N.); kutubuddin.molla@icar.gov.in (K.A.M.); kalyaniaau@gmail.com (K.M.B.); mitadrumukherjee.1@gmail.com (M.M.); agriparames07@gmail.com (P.C.); smitraray@gmail.com (S.S.); rnrao14@yahoo.com (N.R.R.); onsingh01@yahoo.com (O.N.S.)

**Keywords:** differential gene expression, heterosis, hybrid rice, transcriptome, RNA-Seq

## Abstract

RNA-Seq technology was used to analyze the transcriptome of two rice hybrids, Ajay (based on wild-abortive (WA)-cytoplasm) and Rajalaxmi (based on Kalinga-cytoplasm), and their respective parents at the panicle initiation (PI) and grain filling (GF) stages. Around 293 and 302 million high quality paired-end reads of Ajay and Rajalaxmi, respectively, were generated and aligned against the Nipponbare reference genome. Transcriptome profiling of Ajay revealed 2814 and 4819 differentially expressed genes (DEGs) at the PI and GF stages, respectively, as compared to its parents. In the case of Rajalaxmi, 660 and 5264 DEGs were identified at PI and GF stages, respectively. Functionally relevant DEGs were selected for validation through qRT-PCR, which were found to be co-related with the expression patterns to RNA-seq. Kyoto Encyclopedia of Genes and Genomes (KEGG) pathway analysis indicated significant DEGs enriched for energy metabolism pathways, such as photosynthesis, oxidative phosphorylation, and carbon fixation, at the PI stage, while carbohydrate metabolism-related pathways, such as glycolysis and starch and sucrose metabolism, were significantly involved at the GF stage. Many genes involved in energy metabolism exhibited upregulation at the PI stage, whereas the genes involved in carbohydrate biosynthesis had higher expression at the GF stage. The majority of the DEGs were successfully mapped to know yield related rice quantitative trait loci (QTLs). A set of important transcription factors (TFs) was found to be encoded by the identified DEGs. Our results indicated that a complex interplay of several genes in different pathways contributes to higher yield and vigor in rice hybrids.

## 1. Introduction

Heterosis or hybrid vigor refers to the phenomenon of superior performance of a hybrid over its parent in terms of biomass production, development rate, grain yield, and stress tolerance. Utilization of heterosis has tremendously increased productivity of many crops, globally. Since rice (*Oryza sativa* L.) is a staple food crop for more than half of the world’s population, the ability to increase its yield potential would be a key factor in achieving the global rice requirement of 810 million tons by 2025 [1]. The plateauing trend in the yield of HYVs (High-Yielding Varieties), declining and degrading natural resources, like land and water, and the acute shortage of labor make the task of increasing rice production quite challenging. Rice grain yield is a complex agronomic trait determined by three component traits, viz. panicles number/plant, grains number/panicle, and grain weight [2]. Understanding heterosis in relation to these traits contributing to grain yield is expected to help its utilization in rice production.

Dominance [3] and overdominance hypotheses [4,5] were coined a long time ago to explain heterosis. However, the exact molecular mechanism behind heterosis still remains enigmatic. The three line systems of hybrid rice involve use of cytoplasmic male sterile line (CMS, A line) maintained by its isonuclear line (maintainer, B-line) and restorer (R line), in a two-step seed production system: (1) CMS line multiplication from an A × B cross in the field through natural outcrossing and (2) hybrid (A × R) seed production [6]. Advanced molecular techniques, such as the Expressed Sequence Tag (EST) library [7], serial analysis of gene expression (SAGE) [8,9], and microarray [7], have been employed to study genome-wide gene expression profiles of hybrids and their parents. With the advent of new sequencing technologies, it has become much easier to investigate the changes at the transcriptome level. Transcriptome analysis using RNA-Seq is advantageous in several aspects over the other existing methods of genome-wide expression analysis, which could provide insights into the molecular and genetic mechanism of heterosis in rice [10]. Using the RNA-Seq technique, identification of genes which are expressed differentially between rice hybrids and their parents has been performed in earlier studies [11,12,13,14,15]. These studies reported that heterosis in the hybrid is the result of allelic interactions between its parental genomes and subsequent altered programming of genes promoting growth, tolerance to stress, and fitness of hybrids [16]. However, the usage of flag leaf in our study makes our exceptional when compared other reports, as it is considered as the energy sources for rice grain development [17,18,19]. In order to deduce the complex molecular mechanism of rice heterosis, an in-depth understanding is required at the level of global gene expression in hybrids and the parents.

Basically, rice completes two distinct sequential growth stages: vegetative and reproductive. The reproductive stage is subdivided into pre-heading and post-heading periods. The latter is better known as the ripening period. Yield capacity, or the potential size of crop yield, is primarily determined during pre-heading. Ultimate yield, which is based on the amount of starch that fills spikelets, is largely determined during the post-heading stage [20]. Therefore, we aimed at analyzing the global transcriptome of two rice hybrids, Ajay and Rajalaxmi, and their parents at two developmental stages, viz. panicle initiation (PI) and grain filling (GF), using RNA-Seq technology to investigate the differential expression of genes, which might provide a baseline for further detailed studies for exploiting heterosis in rice.

Ajay and Rajalaxmi were developed using CRMS31A (wild-abortive (WA)-cytoplasm based CMS line) and CRMS32A (Kalinga I- cytoplasm-based CMS line), respectively, which are phenotypically different from each other. The same fertility restorer line (male parent, IR 42266-29-3R) with different CMS lines (female parent) was used for both the hybrids. It is noteworthy that the Rajalaxmi hybrid has been found to be tolerant of abiotic stresses, like salt and cold, at the germination/seedling stage.

## 2. Results

### 2.1. Differential Gene Expression among Hybrids and Its Parents

RNA-Seq generated 331 million short reads and 66 Gb original data for the Ajay group, whereas it generated 341 million short reads and 68 Gb for Rajalaxmi (Table 1 and Table 2). After discarding adaptors and low quality reads, 293 and 302 million high quality reads, representing 88.5% of both the original raw reads, were selected for downstream analysis in Ajay and Rajalaxmi, respectively. Transcriptome profiles of both the hybrids, Ajay and Rajalaxmi, were found to be distinct as compared to their respective parental lines at the PI stage (additional Figure 1a). In contrast, the transcriptome profile of both the hybrids showed a closer relationship with its male parental line (IR-42266-29-3R) at the GF stage (additional Figure 1b). These observations are consistent with morphological observations, where the hybrids were dissimilar from either of the parents at the PI stage and similar to the male parent at the GF stage phenotypically. The observation at the GF stage is also consistent with the results from the hierarchical cluster analysis.

As compared to its parents, 2814 and 4819 differentially expressed genes (DEGs) were present at the PI and GF stages, respectively, in Ajay (Appendix A; Figure 1a). Out of 2814 DEG_S_ at the PI stage, 447 transcripts (15.87%) showed differences between Ajay and CRMS31A, in which 98 (3.5%) were up-regulated and 349 (12.4%) were down-regulated. While 2529 transcripts (89.78%) exhibited differences between Ajay and IR-42266-29-3R, 187 (6.65%) were up-regulated and 2342 (83.22%) were down-regulated in Ajay. On the other hand, at the GF stage, 3747 transcripts (77.75%) showed differences in expression between Ajay and CRMS31A, among which 1302 (27%) were up-regulated and 2445 (50.73%) were down-regulated, whereas 1667 transcripts (34.60%) exhibited differences between Ajay and IR-42266-29-3R, of which 699 (14.50%) and 968 (20.1%) were found to be up- and down-regulated, respectively, in Ajay (Table 3).

As compared to Ajay, Rajalaxmi showed a lower number (660) of transcripts at PI and a slightly higher number (5264) of transcripts at GF and its parents (Appendix A; Figure 1b). Out of the 660 DEGs at the PI stage, 268 transcripts displayed a difference between CRMS32A and Rajalaxmi, in which 145 (24%) and 123 (20.5%) DEGs were up-regulated and down-regulated, respectively, while 559 transcripts expressed differentially between Rajalaxmi and its male parent, with 153 (25.5%) up-regulated and 406 (76.67%) down-regulated transcripts in Rajalaxmi. Similarly, at the GF stage, out of 5264 DEGs, 3963 showed differences between Rajalaxmi and its female parent, whereas 1659 exhibited differences between the hybrid and its male parent (Table 3).

### 2.2. Functional Classes, Pathways Enriched, and Key Genes Involved in Heterosis

In Ajay, out of total 7633 DEGs, 6651 were Gene Ontology (GO) term enriched, of which 2846 DEGs were at the PI stage and 4165 DEGs were at the GF stage (Appendix A). Similarly, in Rajalaxmi, out of 5924 DEGs, 4991 were GO term enriched, among which 480 DEGs were at the PI stage and 4511 DEGs were at the GF stage (Appendix A). The GO enriched unigenes were classified into three main functional categories, viz. molecular function, cellular component, and biological process (Figure 2a,b). In both the hybrids, considering the cellular component category, cell, cell parts, and organelle, all were prominently represented, while binding and catalytic process dominated the molecular function category.

Furthermore, over-represented (*p* < 0.05) the GO term enriched DEGs in the biological process category were analyzed in both the hybrids (Table 4). In this analysis, carbohydrate metabolic process, establishment of localization, generation, and precursor metabolites and energy, response to biotic stimulus, and transport and localization were found to be significantly enriched in the GO term in both developmental stages of both the hybrids (Appendix A). The result indicated that those processes might play significant role in yield enhancement of hybrid rice. However, protein modification, regulation of biological and cellular processes, response to stress, and signal transduction DEGs were only enriched at the PI stage in Ajay, where tropism was enriched in the GF stage. In the case of Rajalaxmi, cellular biosynthetic process and macromolecule biosynthesis process were enriched in both the PI and GF stages, whereas biosynthetic process and gene expression were only enriched in the GF stage. These results suggest that, in both stages, some of the different biological processes were required in response to specific stimuli.

Carbohydrates, energy, lipid metabolism, and biosynthesis of secondary metabolism and amino acid metabolism were found to be the major Kyoto Encyclopedia of Genes and Genomes (KEGG) pathways in both the hybrids at the PI and GF stages (Figure 3 and Figure 4). More interestingly, the KEGG orthology term related to all categories of functional classification in the GF stage was comparatively more enriched in DEGs than the PI stages in both the hybrids, except energy metabolism and signal transduction in Ajay. Under biological process, the highest number of DEGs found were those involved in energy metabolism, i.e., photosynthesis, oxidative phosphorylation, carbon fixation, methane, and nitrogen metabolism, at the PI stage, while at the GF stage, carbohydrate metabolism, i.e., glycolysis, TCA cycle, PP pathway, inositol phosphate, butanoate, propanoate, pyruvate, fructose, mannose, galactose, ascorbate, aldarate, amino sugar, nucleotide sugar, glyoxylate, dicarboxylate, and starch and sucrose metabolism were enriched (Appendix A).

The different families of transcription factors (TFs) play an extensively critical role in controlling numerous metabolic pathways, phenotypes, and ultimately traits. Among different TFs, the ethylene responsive transcription factor (ERF) family plays a vital role in plant growth and enables plants to fight different stresses. In the present study, ERF encoding DEGs were enriched more at the GF stage than at the PI in both the hybrids (Appendix A). It is clear from the result that the ethylene signaling pathway is in operation at the GF stage in a higher rate than PI.

The MYB family is another important family, and it is one of the abundant groups of TFs in plants that plays significant role in primary and secondary metabolism, development, and stress responses. A trend in enrichment of MYB/MYB-like TF encoding DEGs showing more number (39 and 38 for Ajay and Rajalaxmi, respectively) at the GF stage than the PI (15 and 6, respectively) was observed in our study. The MYB family TF (homeodomain-like, containing protein genes, Os03g0764600) was highly expressed in both the hybrids as compared to their CMS lines.

Among the other TF families, WRKY, bZIP, NAC, and bHLH were found to be in higher number as compared to all other TF families (Figure 5). Following a common trend, more DEGs coding for TFs was observed at the GF stage than the PI, except for WRKY in Ajay, which showed 58 TFs at the PI stage and 57 TFs at the GF stage. Commonly expressed bZIP TFs in both the hybrids in the GF stage were analyzed, and most of them were found to be up-regulated in hybrids, either for both the parents or for one of the parents. OsbHLH065 (Os04g41570, basic/Helix-Loop-Helix TF) was up-regulated in both the hybrids as compared to parental lines. Another bZIP TF, the actin-binding FH2 domain containing protein, (Os12g0105300) also up-regulated in both hybrids over both the parents.

### 2.3. Validation of DEGs Identified in RNA-Seq Data Using qRT-PCR

Of all the DEGs identified in hybrids and parental lines using RNA-Seq analysis, 40 DEGs were selected for validation through qPCR analysis in both the PI and GF stages. The list of selected DEGs specific to each combination of hybrid versus A line and hybrid versus R lines is given in Appendix A. In general, the pattern and fold change (FC) in the expression of genes was similar between the RNA-Seq data and qPCR analysis (Figure 6).

### 2.4. Yield-Related Quantitative Trait Loci (QTLs) Identification by Mapping of DEGs

Since the superiority of hybrid is mainly considered for yield over its parents, the DEGs were mapped with the yield related QTLs. Among the total 7633 DEGs in Ajay, 7341 (96.1%) were mapped on yield-related QTLs, while 5814 (98.14%) of total 5924 DEGs of Rajalaxmi were mapped. Many of these are well characterized with regard to traits, such as 1000-seed weight, filled grain number, grain number, and grain yield per panicle. Consequently, the entirety of the mapped DEGs was categorized into four groups, i.e., grain weight, grain number, yield per panicle/plant, and other traits related to yield (Appendix A). In Ajay, most of the QTLs and DEGs were involved in grain/flower/panicle number (43.2%), followed by biomass/grain yield (26.4%), seed/grain weight (21.8%), and other yield-related traits (8.6%) at the PI stage, while at the GF stage, QTLs were involved in grain/flower/panicle number (43%), biomass/grain yield (27%), seed/grain weight (21.5%), and other yield-related traits (8.5%). In the case of Rajalaxmi, most of the yield-related QTLs and DEGs were involved in grain/flower/panicle number (PI, 45%; GF, 43%), followed by biomass/grain yield (PI, 25%; GF, 26%), then seed/grain weight (PI, 22%; GF, 21%), and other yield-related traits (PI, 9%; GF, 9%) (Appendix A). Since Rajalaxmi is reported to be higher in yield, along with salinity and cold tolerance (at seedling stage), the correlation of QTL related to abiotic stress with the DEGs were investigated for Rajalaxmi. A total of 5056 DEGs (85.34%) were mapped on abiotic stress-related QTLs (Supplementary file 4.3). The percentage of DEGs mapped QTLs for susceptibility was higher in the GF stage than the PI stage, which is in accordance with its ability of tolerance to salinity and cold stress at the early stage (Appendix A). 

Further, DEGs could also be mapped to QTLs of small intervals (spanning no more than 100 genes). In the case of Ajay, 241 DEGs (3.15%) were located in 142 yield-related QTL of small intervals at the PI stage, while at the GF stage, 430 DEGs (5.63%) were located in 180 yield-related QTL of small intervals. Further, comparison of the PI and GI stages revealed that 96 DEGs (1.25%) were located in 78 QTL of small interval in both the stages. In Rajalaxmi, 21 DEGs were found to be located in 38 yield-related QTL of small intervals at the PI stage, while at the GF stage, 400 DEGs were found to be positioned in 187 yield-related QTL of small intervals.

## 3. Discussion

The exploitation of heterosis has been one of the most notable contributions of genetics to agriculture, enhancing the vigor in hybrid crop, as well as in livestock. However, the genetic and molecular mechanism underlying heterosis is undoubtedly complicated and has been subjected for intense research and speculation for over a century. Even after several genetic hypotheses of heterosis have emerged, such as the ‘‘dominance’’ and ‘‘overdominance’’, the basic molecular mechanisms remain clouded [21,22,23]. According to a recent epistasis hypothesis, interactions of alleles at different loci from two parents are responsible for heterosis in the F_1_ hybrid [24,25]. Since revolution in sequencing technology allows investigation at the transcriptome level, we aimed at employing RNA-seq to reveal various facets associated with heterosis in two rice hybrids, Ajay and Rajalaxmi, and their respective parents. We selected the PI stage, the first stage in the reproductive phase of growth, denoting the commencement of panicle formation, including male and female organs, and the GF stage, a highly coordinated developmental process, involving synthesis of a large amount of storage compounds and transportation into the rice endosperm. These two stages are considered to be very crucial for determining yield.

It was found that there were significantly more DEGs in both the hybrids at the GF stage (4819 for Ajay and 5264 for Rajalaxmi) than at the PI stage (2814 for Ajay and 660 for Rajalaxmi). As the GF stage is more metabolically active than the PI stage in respect of yield of the hybrid, it can be speculated that there are more differences at the transcriptome level of hybrid rice over their parents. Our results are in consonance with the assumption and further encourage speculation that DEGs at the GF stage might be the pivotal role player in heterosis. The study of Zhai et al. [12] on rice root heterosis demonstrated the accumulation of more differentially expressed transcripts in the heading stage than in the tillering stage. As the male parent was common for both the hybrids, the resultant variation is expected because female parents have different nuclear, as well as cytoplasmic, backgrounds. In the biological process category, cellular and metabolic process were found to represent the most abundant class of DEGs, indicating extensive metabolic activities that were taking place in leaves of hybrid plants during both the stages. In addition, significant DEGs involved in developmental processes might respond to stimulus, localization, pigmentation, and the reproductive process of both these stages.

There are considerable differences in the physiological process between these two stages and the carbohydrate metabolic pathway, which is dominated in heterosis, followed by amino acid metabolism, signal transduction, metabolism of cofactors and vitamins, and energy metabolism and translocation. Variations in terms of the number and type of DEGs between two hybrids observed in terms of gene expression profiling might be due to different female lines used for the development of Ajay and Rajalaxmi.

### 3.1. Photosynthesis and Yield in Hybrid Rice

Many biological, as well as physiological, factors, such as carbon dioxide (CO_2_) fixation, conversion of primary photosynthates into sugar, and their deposition in the form of starch grain, determine the productivity (biomass) of rice plant. Photosynthesis is the basic pathway for carbohydrate (glucose) synthesis by fixing carbon dioxide with the help of photogenerated NADPH (nicotinamide adenine dinucleotide phosphate hydrogen) and ATP (adenosine triphosphate). Functional analysis of hybrid transcriptomes indicated that a large number of DEGs were involved in multiple pathways, including photosynthesis, carbohydrate metabolism, energy metabolism, and generation of precursor metabolites and energy, in both the developmental stages. However, significant DEG enrichment was not observed for photosynthesis at the PI stage of Rajalaxmi. For yield determination, the PI stage may be regarded as a preparatory phase that is more energy demanding than the GF stage. So, it can be assumed that higher operation of the pathway related to energy generation at the PI stage might be one of the reasons for yield advantage in hybrids over their parents. Interestingly, our results showed that the pathway related to generation of energy was significantly more enriched with DEGs in both the hybrids at the PI stage than the GF stage against that of their parents. The analyzed data showed that many genes involved in photosynthesis were up-regulated at the PI stage in both the hybrids as compared with their parents, which is in accordance with a previous study [8]. Many recent studies demonstrated a close relationship between enhanced photosynthesis, biomass, and yield, suggesting that increase in photosynthesis causes enhancement of yield when other genetic factors are not altered [26]. The DEGs encoding photosystem I &II, F-type ATPase, ferredoxin-NADP reductase, and cytochrome b6-f complex were enriched, which play important roles in photosynthetic electron transfer in thylakoids. Moreover, NAD(P)H-quinoneoxidoreductase (ndhI), ATP synthesis coupled electron transport genes were also up-regulated in hybrids over both the parental lines, suggesting higher ATP generation in hybrids that could have ultimately boosted up various energy consuming pathways.

In the carbon-fixation pathway, many DEGs exhibited higher expression in both the hybrids from one of the parental lines directly related to yield, e.g., glyceraldehyde-3-phosphate dehydrogenase (GAPDH), Ribulose-Bisphosphate Carboxylase (RuBisCO), malate dehydrogenase (MDH), pyruvate orthophosphate dikinase (PPDK), and fructose-1,6- bisphosphatase (FBP). For instance, RuBisCO (Ribulose-1,5-Bisphosphate Carboxylase/Oxygenase) is the key enzyme for CO_2_ fixation in a rice plant and subsequent formation of starch and sucrose [27]. The same enzyme is also responsible for photorespiratory loss of CO_2_. Interestingly, a RuBisCO large chain precursor was down-regulated in the hybrid Ajay more than the female parent at the PI stage. This might be due to the higher effort of reduced photorespiratory loss in the hybrid than its parent. This analysis revealed that DEGs involved in the carbohydrate metabolism pathway had a larger fraction of up-regulated genes than down-regulated ones [21,28].

### 3.2. Role of Carbohydrate Metabolism in Heterosis

Besides photosynthesis, sucrose and starch pathways, oxidative phosphorylation, and TCA cycle may also play contributory role to heterosis [29]. In our study, more DEGs were enriched in glycolysis, TCA cycle, pentose phosphate pathway, and starch and sucrose metabolism at the GF stage than at the PI stage in both the hybrids (Appendix A). Comparatively, the oxidative phosphorylation pathway was enriched by more DEGs at the PI than the GF stage for Ajay, whereas the result was reversed in the case of Rajalaxmi, which needs further examination.

It has been noticed that genes involved in carbohydrate biosynthesis have a much higher expression in the GF stage than the PI stage. Interestingly, genes involved in TCA cycle, starch and sucrose metabolism, amino sugar and nucleotide sugar metabolism, ascorbate and aldarate metabolism, and glyoxylate and dicarboxylate metabolisms pathways were down-regulated in the PI stage, while they were found to be up-regulated in the GF stage. Some other enzymes also involved in a different metabolic pathway related to carbohydrates are malate dehydrogenase (MDH), mannose-6-phosphate isomerase (MPI), UDP-glucose 4-epimerase (GALE), beta-glucosidase (GBA), starch synthase, and sucrose-phosphate synthase, which were up-regulated in both the hybrids compared than their parents, which is corroborated with the findings of Zhai et al. [12]. However, galactokinase (GALK1) and alpha-galactosidase (GLA) genes were down-regulated in hybrids as compared to CMS lines. These results are in agreement with the fact that starch biosynthesis in the PI stage is lower than in the GF stage in a hybrid. In this study, AAA-type ATPase family protein genes were highly up-regulated in both the hybrids as compared to one of the parental lines. AAA proteins coupled with chemical energy provided by ATP hydrolysis to conformational changes are transduced into mechanical force exerted on a macromolecular substrate [30].

### 3.3. Stress Tolerance Potential of the Hybrids

In consonance with the cold and salinity resistance characters of the hybrid, Rajalaxmi, we found many DEGs were located on abiotic stress-related QTLs. The glutathione metabolic pathway is well known to prevent cellular and organellar damage from reactive oxygen species (ROS) generated from different stresses. Results of the KEGG pathway analysis of DEGs revealed the involvement of the glutathione metabolism at the GF stage in both the hybrids, and the genes of the pathway also showed up-regulation in the hybrids over their female parent. Abscisic acid (ABA) and ethylene are two stress-responsive phytohormones that were found in the enrichment of the signaling pathway. Several stress responsive TF encoding DEGs have been found to exhibit higher expression in hybrids than their parents (Appendix A).

Plant secondary metabolites are crucial for the plant in adapting to an adverse environment for defense [31]. A large number of DEGs were assigned to the pathway related to various secondary metabolite syntheses, indicating their role in making the hybrids better-suited to the environment [11,32]. Similarly, early light-induced protein (ELIP) and chloroplast precursor genes were also highly up-regulated in both the hybrids at the GF stage. The ELIPs belong to the multigenic family of light-harvesting complexes that bind chlorophyll and absorb solar energy in green plants. ELIPs accumulate transiently in plants exposed to high light intensities and protect the plants from photooxidative stress, as reported in Arabidopsis [33], thereby promoting thermo-tolerance in plants. The type and pattern of expression of DEGs in the hybrids described above indicate that a large number of stress responsive molecular functions play a significant role in the hybrid over their parents. A similar kind of result was demonstrated in the hybrid rice LYP9 in a previous study [13].

### 3.4. Signal Transduction

Plant hormones, especially ethylene and ABA, play important roles in regulating GF [34]. In our study, 48 DEGs in Ajay and 47 in Rajalaxmi at the GF stage were found to be involved in phytohormone signaling (Appendix A). In particular, the role of ABA in GF was found to be complicated, and the ABA responsive element binding factor (Os06g10880) was down-regulated in hybrids from both the parental lines, reducing the possibility of activation of ABA-induced signaling pathways. High ABA signaling at the GF stage is detrimental to yield as a high concentration of ABA reduces transport of sucrose into the grains and lowers the ability of grains to synthesize more starch [35,36], while the appropriate concentration of ABA can enhance sucrose synthase (SUS) activity [37], increase the expression of other genes related to starch metabolism [38], and achieve higher grain yield [39].

A higher level of ethylene signaling in developing seeds correlates negatively with starch metabolism-related enzyme activities and always leads to poor GF [28,40,41]. Over-expression of the ethylene receptor gene (ETR2) in rice down-regulates the monosaccharide transporter gene and, consequently, prevents sugar translocation from the stem to grains, leading to reduced grain weight [42]. In consonance with those studies, our result revealed the down-regulation of ethylene receptor genes (ETR) (Os04g08740, Os02g57530) in both the hybrids over their CMS lines.

The genes encoding ZIM (zinc-finger inflorescence meristem) domain containing protein (Os03g0180800, Os03g0180900, Os03g0181100, Os10g0392400, and Os01g0808100) were down-regulated in both the hybrids as compared to their male parent (R line). Jasmonate ZIM-domain (JAZ) proteins act as repressors of jasmonate (JA) signaling [43]. As degradation of JAZ1 is necessary during spikelet development in rice [44], the down-regulation of JAZ genes in a hybrid may be attributed to more efficient spikelet formation than its male parent.

### 3.5. DEGs Associated with Yield-Related QTL

The economically important traits of crop plants are quantitative in nature, including grain yield of rice. In this study, we investigated the association or relationship between DEG, QTL, and heterosis. Mapping of all DEGs to known yield-related QTLs revealed that the majority (98.03%) of DEGs were located in the interval of QTLs, which is in agreement with the results of Song et al. [9] and Wei et al. [11], who mapped 97.9% and 85.5% DEGs, respectively, to the known QTLs out of total DEGs in hybrids. We could map few DEGs of the PI stage to yield-related QTL in small intervals in both the hybrids; to mention a few important ones, PSII (Photosystem II) polypeptide subunits, which are responsible for photosynthesis, the raffinose synthase family protein for carbohydrate metabolism, blight-associated protein p12 for cell wall modification, and prephenate dehydratase domain containing protein for amino acid metabolism are the DEGs mapped to small interval QTL. A similar kind of result was obtained in an earlier study [11]. The five most significant DEGs-enriched pathways were glycolysis, photosynthesis, plant hormone signal transduction, and starch and sucrose metabolism. Interestingly, the important DEGs of each pathways, viz. glycolysis (6-phosphofructokinase, GAPDH, and aldose 1-epimerase), photosynthesis (cytochrome b6-f complex and photosystem I, II), phytohormone signaling pathway (stem-specific protein TSJT1, flavonol synthase/flavanone 3-hydroxylase), and starch and sucrose metabolism, such as sucrose-phosphate synthase, sucrose synthase, beta-glucosidase, and glucose-1-phosphate adenyl-transferase, were found to be located at important yield-related QTL regions. According to a previous report, genes involved in photosynthesis were found to be located at yield-related QTLs [9]. Interestingly, major validated yield-related rice QTLs, like *qSPP1, qSPP3, qspp8*, and *gw3.1* [2], were found to harbor our collection of DEGs. Furthermore, in consonance with a previous report [45], three DEGs at the PI stage and nine DEGs at the GF stage were found to be located at a well-known QTL (AQGK001) site related to grain productivity in rice. These results suggest that the DEGs between a hybrid and its parents most likely played a significant contributory role in heterosis.

### 3.6. TFs and Their Role in Heterosis

Since there is no report available for the genes regulated by TFs for heterosis and their exact functions, our result showed a diverse array of TFs to be functional in the expression level of a large number of genes that might be playing significant role in heterosis. A previous study in rice hybrids suggested that differentially expressed TFs and polymorphism in promoter elements are the two combinatorial role players in rice heterosis [46].

Among different TFs, the ERF family plays a vital role in plant growth and enables plants to fight different stresses [47,48,49,50]. ERF-encoding DEGs were enriched more at the GF stage than the PI stage in both the hybrids. It is clear from the result that the ethylene signaling pathway is in operation at the GF stage at a higher rate than at the PI stage.

The MYB family is involved in plant development, secondary metabolism and biotic and abiotic stress responses [51,52]. We found a similar trend of enrichment of MYB/MYB-like TF encoding DEGs showing higher numbers (39 and 38 for Ajay and Rajalaxmi, respectively) at the GF stage than the PI stage (15 and 6, respectively) as compared to its parents. Simultaneously, the MYB family TF (homeodomain-like containing protein genes, Os03g0764600) was highly expressed in both the hybrids as against of their CMS lines.

Among the TF family, in a descending order WRKY, bZIP, NAC, and bHLH were found to be more abundant than the others. Following a common trend, more DEGs coding for TF was observed at the GF stage than the PI stage, except for WRKY in Ajay showing 58 at the PI stage and 57 at the GF stage. Commonly expressed bZIP TFs in both the hybrids in the GF stage were analyzed, and most of them were found to be up-regulated in hybrids. The gene *OsbHLH065* (Os04g41570, basic/Helix-Loop-Helix TF) was up-regulated in both the hybrids as compared to the parental lines. Another bZIP TF, the actin-binding FH2 domain containing protein (*Os12g0105300*) also up-regulated over both the parents in the two hybrids. However, very little is known about the genes regulated by TFs and their exact functions in rice plant metabolism. Therefore, it can be assumed from our result that a diverse array of TFs is functional to fine tune the expression level of a large number of genes which might be playing a significant role in heterosis.

## 4. Materials and Methods

### 4.1. Plant Materials

Two popular rice hybrids, viz. Ajay and Rajalaxmi, developed at ICAR-NRRI and their parents CRMS31A (female for Ajay), CRMS32A (female for Rajalaxmi), and IR-42266-29-3R (common male for both) were used as the source of plant material in the present study. Seeds were sown during *Kharif* in 2013 at experimental farm of ICAR-National Rice Research Institute, Cuttack. Twenty-one day-old seedlings were transplanted into experimental plots following the agronomic practices recommended for rice in six replications. PI (when a young panicle has grown about 1 mm long and can be seen with the naked eye or through a magnifying glass) and GF (when milky accumulation is noticeable inside florets) stages were determined using standard methods [20]. Flag leaf samples at two different stages, PI at stage 1 and GF at stage 2, were harvested from all rice lines in three replications, snap-chilled in liquid nitrogen, and immediately stored at −80 °C for further use.

### 4.2. RNA-Seq and Differential Gene Expression

The total RNA was extracted from 100 mg of leaf samples at both the stages using a XcelGen Plant RNA mini prep kit (Xcelris Genomics, Ahmedabad, Gujarat, India), following the manufacturer’s instruction, from three replications and pooled. The yield and purity of RNA were assessed by determining the ratio of 260/280 nm. Further, RNA quality was checked using a RNA 6000 Nano chip (Agilent Bioanalyzer 2100 Santa Clara, CA, USA). Samples possessing RIN (RNA Integrity Number) values >8.5 were used. Sequencing libraries were prepared using an Illumina TruSeq^®^ RNA Library Preparation Kit (Illumina^®^, San Diego, CA, USA) as per manufacturer’s protocol and run on an Illumina HiSeq2000.

The generated raw reads were filtered for low quality bases (Q < 25) and adaptor sequences, which were back mapped with the rice reference genome of *Oryza sativa* japonica group cultivar of Nipponbare (MSU 7.0, East Lansing, MI, USA). For the identification of DEGs (Differential Expressed Genes) between the hybrids and parental lines and to understand the genes related to heterosis, differential gene expression was compared among the A line, R line, and the resultant hybrids in both the PI and GF developmental stages. Analysis of functional categories of DEGs and their expression pattern revealed multiple pathways and a complex mode of regulation involved in heterosis. The differential expression was analyzed using Tophat v1.3.3 and Bowtie v0.12.9 (Baltimore, MD, USA). The expression level of each gene was quantified in terms of fragments per kilobase of exon per million fragments mapped (FPKM), calculated false discovery rate (FDR), and estimated fold change (FC) and log2 values of FC. Transcripts with FDR ≤ 0.05 and estimated absolute log_2_ (FC) ≥ 1 were considered to be significant in differential expression. In order to investigate correlations among the transcriptome profiles of the hybrids and their respective parents at two different stages, a cluster analysis method was used, employing cluster 3.0 software, Tokyo, Japan.

All predicted DEGs were assigned to different functional categories using a web-based tool and database "AgriGO" [53] and further annotations were plotted using Web Gene Ontology Annotation Plot (WEGO) software [54]. The KEGG pathway database was used to identify metabolic pathways in which DEGs are involved [55].

### 4.3. RNA Extraction and Real-Time Quantitative PCR (qRT-PCR)

The total RNA of flag leaf samples of three biological replicates at the PI and GF stages from hybrid rice, CMS lines, and the restorer line was isolated as described earlier and utilized for cDNA synthesis. The quality of the isolated RNA was checked by Nanodrop UV-visible Spectrophotometer (Thermo Scientific, Wiggins Ave, Bedford, MA, USA) and 1.5% agarose gel. Then, 5 µg of total RNA was treated with DNase I (NEB, USA) enzyme to remove the DNA contamination. Further, 2 µg of DNase I treated RNA was used for first strand cDNA synthesis using qScript^TM^cDNASupermix (Quanta bioscience, Gaithersburg, MD, USA). A total of 40 selected DEGs were used in qRT-PCR analysis for validation of RNA-Seq data (Appendix A). Quantitative-PCR (qRT-PCR) for the expression of the selected DEGs was performed using Mastercycler Realplex (Eppendorf, Germany). Rice 25S rRNA was used as reference gene for cDNA normalization. For q-PCR analysis, three biological and three technical replicates for each sample were used to identify the expression pattern of the DEGs. SYBR, Premix Ex TaqII, and Takara were used for the PCR cycle with the following cycle parameters: initial denaturation of 95 °C for 30 s, followed by 40 cycles of 95 °C for 5 s denaturation, and 60 °C for 30 s of combined annealing and extension followed by melting curve analysis. The specificity of the amplicons was analyzed through melt curve and ∆∆*^C^*^T^ method was used for FC expression analysis. The relative FC in hybrids with respect to restorer and CMS lines was calculated by considering the expression in both the lines as one and compared with hybrids. The details of primer pairs used for the qRT-PCR are depicted in Appendix A.

### 4.4. In-Silico Mapping the DEGs with the Known QTLs

The DEGs of both the hybrids and parental lines, were mapped with the physical positions of rice QTLs available at TIGR (The Institute for Genomic Research) release 5 genome, Gramene (www.gramene.org) for abiotic stress, anatomy, biochemical, biotic stress, development, quality, sterility, fertility, vigor, and yield deposited. For better demonstration of the relationship between DEGs and QTL, yield-related QTLs were classified according to the number of genes in each chromosome region, and an enrichment test was performed according to the method described earlier [56].

### 4.5. TF Identification

The DEGs were searched against the rice TF database (PlantTFDB v2.0, Haidian, China) using BLASTX with an E-value cut-off of 1E-05 for the identification of TF families represented.

## 5. Conclusions

In conclusion, our study presents the identification of DEGs in two rice hybrids, Ajay and Rajalaxmi, with respect to their parental lines at two distinct developmental stages, viz. PI and GF, in order to better understand the mechanism of rice heterosis. A majority of the DEGs were not only mapped in the yield-related rice QTLs reported earlier, but a large number of TFs in the DEGs were also identified. Differential involvement of several important pathways at two different stages (PI and GF) was unraveled, which enriched our understanding of heterotic gene expression over the developmental lane of hybrid rice. These results could provide a valuable resource to interpret and decipher the complex cellular and molecular mechanism of heterosis in rice.

## 6. Deposited Data

The RNA-seq datasets generated in this study have been submitted to NCBI Sequence Read Archive (SRA) with the Bioproject numberSUB830713 from SRS886677 to SRS1068799.

## Figures and Tables

**Figure 1 ijms-21-01080-f001:**
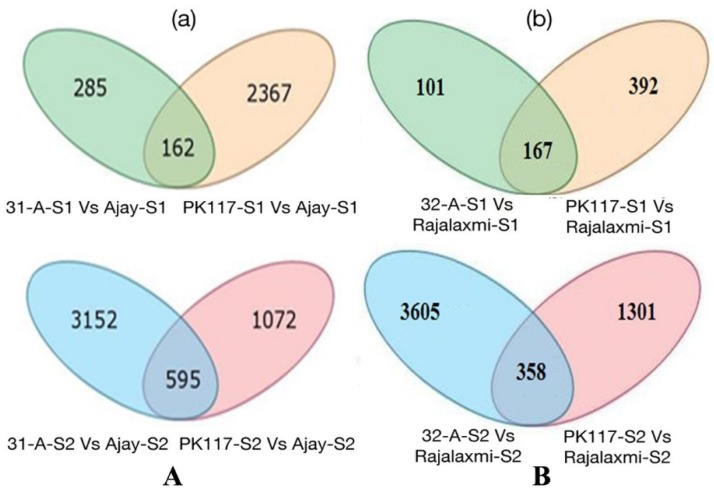
Differentially expressed genes (DEGs) in the rice two heterotic crosses (**A**) Ajay and (**B**) Rajalaxmi. Venn diagram of DEGs between hybrid and their parents at (**a**) panicle initiation (S1) and (**b**) grain filling (S2), PK117: IR42266-29-3R; 31A: CRMS31A; 32A: CRMS32A.

**Figure 2 ijms-21-01080-f002:**
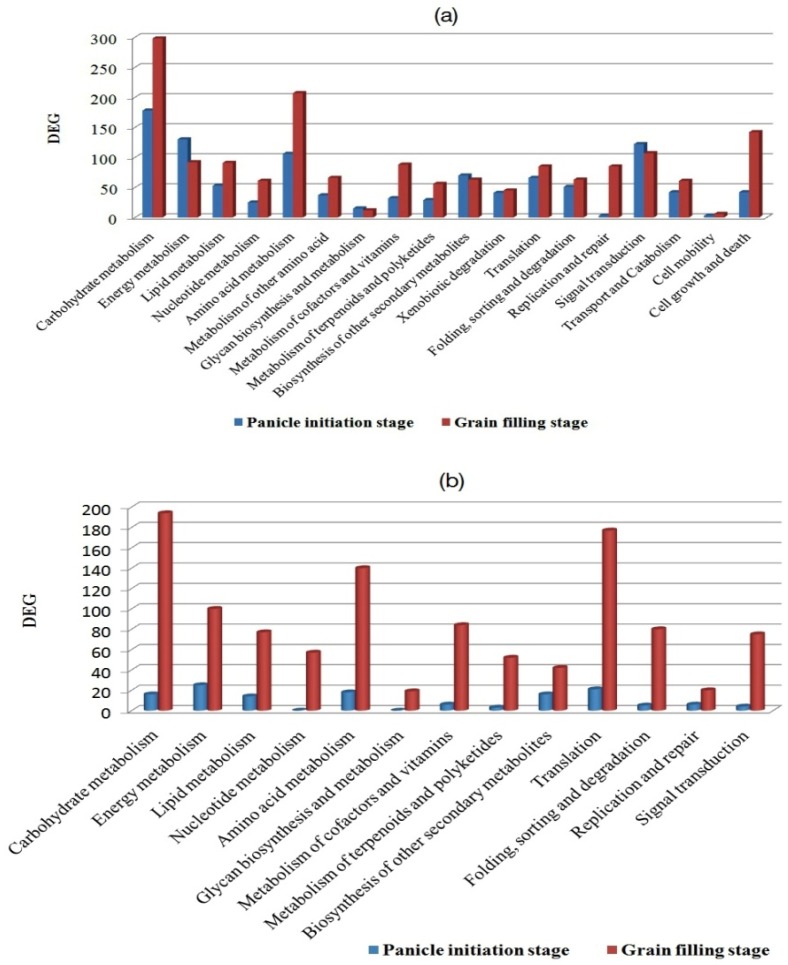
Comparison of Gene Ontology (GO) functional sub-categories of panicle initiation and grain filling stage in two rice hybrids (**a**) Ajay and (**b**) Rajalaxmi.

**Figure 3 ijms-21-01080-f003:**
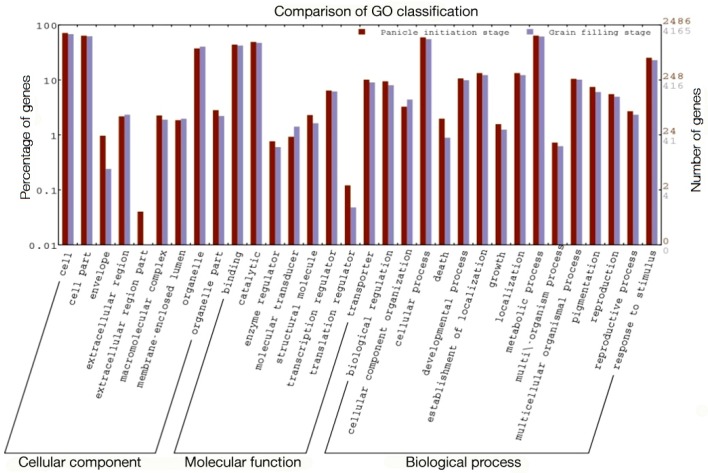
Kyoto Encyclopedia of Genes and Genomes (KEGG) pathway incident of panicle initiation and grain filling stage in Ajay hybrid.

**Figure 4 ijms-21-01080-f004:**
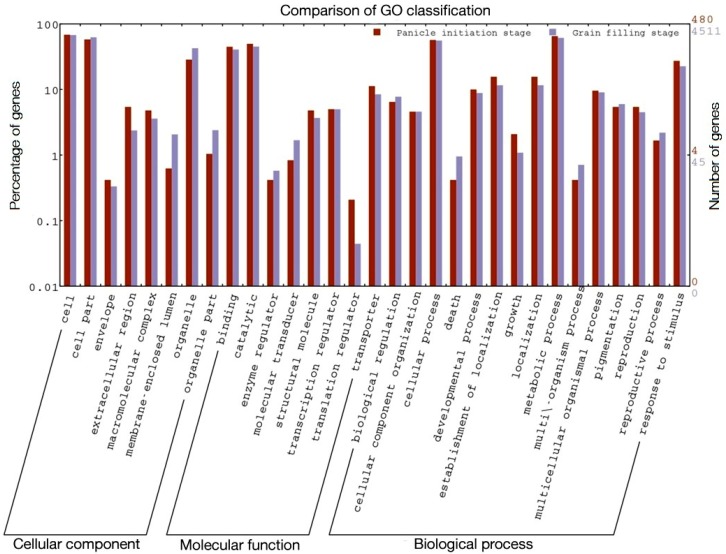
KEGG pathway incident of panicle initiation and grain filling stage in Rajalaxmi hybrid.

**Figure 5 ijms-21-01080-f005:**
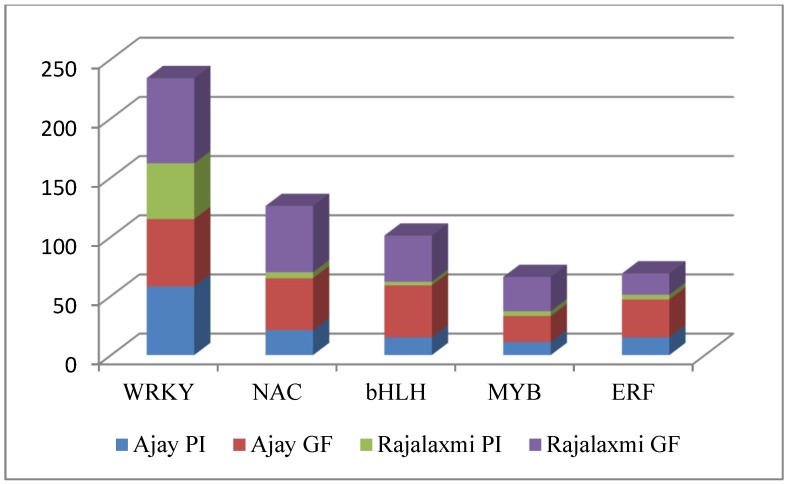
Highly expressed classes of transcription factors (TFs) in two rice hybrids, Ajay and Rajalaxmi.

**Figure 6 ijms-21-01080-f006:**
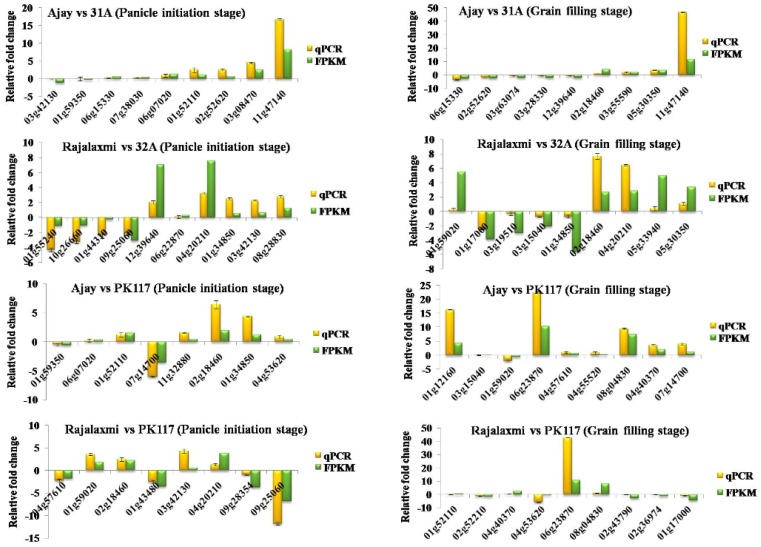
Relative fold change (FC) (qRT-PCR) of selected DEGs identified between rice hybrids (Ajay and Rajalaxmi) and their parental lines at panicle initiation and grain filling stages. FPKM = fragments per kilobase of exon per million fragments mapped.

**Table 1 ijms-21-01080-t001:** A total number of reads mapped with percentage of both panicle and grain filling stage in Ajay and its parental lines.

Samples	Number of Reads Mapped on			Total Number of Reads Mapped	Percentage		
	Exon	Intron	Intergenic		Exon	Intron	Intergenic
CRMS31A-PI	40,218,002	17,052	10,391,484	51,094,480	78.7	0.0	20.3
CRMS31A-GF	65,833,226	87,087	8,165,337	75,026,355	87.7	0.1	10.9
PK117-PI	74,956,596	110,096	10,927,257	87,111,469	86.0	0.1	12.5
PK117-GF	40,239,045	27,816	13,489,140	54,046,852	74.5	0.1	25.0
Ajay-PI	31,177,458	45,592	5,292,987	36,815,969	84.7	0.1	14.4
Ajay-GF	41,685,048	52,409	7,598,708	49,937,045	83.5	0.1	15.2

PI and GF denote panicle initiation (PI) and grain filling (GF) stage.

**Table 2 ijms-21-01080-t002:** A total number of reads mapped with percentage of both panicle and grain filling stage in Rajalaxmi and its parental lines.

Samples	Number of Reads Mapped on			Total Number of Reads Mapped	Percentage		
	Exon	Intron	Intergenic		Exon	Intron	Intergenic
CRMS32A-PI	44,197,542	45,300	11,310,908	56,043,252	78.8	0.08	20.1
CRMS32A-GF	51,792,582	94,467	7,197,864	60,231,007	85.9	0.1	11.9
PK117-PI	74,956,596	110,096	10,927,257	87,111,469	86.0	0.1	12.5
PK117-GF	40,239,045	27,816	13,489,140	54,046,852	74.5	0.1	25.0
Rajalaxmi-PI	33,076,827	95,031	6,658,553	41,603,392	79.5	0.2	16.0
Rajalaxmi-GF	53,353,594	96,355	7,888,639	62,344,409	85.5	1.5	12.6

PI and GF denote panicle initiation (PI) and grain filling (GF) stage.

**Table 3 ijms-21-01080-t003:** Number and classification of DEGs of panicle initiation and grain filling stage.

STAGE	DEG_PP_	DEG_HP_	A/H	R/H	Total
Panicle Initiation (PI) of Ajay	447	2814	447	2529	6237
Grain filling (GF) of Ajay	3924	4819	3747	1667	14,157
Panicle Initiation (PI) of Rajalaxmi	1375	660	268	559	2862
Grain filling (GF) of Rajalaxmi	3582	5264	3963	1659	14,468

R, H, and A refer to PK117 (IR-42266-29-3R), Ajay/Rajalaxmi, and CRMS31A/CRMS32A, respectively. DEG_PP_: DEG between the parents; DEG_HP_: DEG between the hybrid and its parents.

**Table 4 ijms-21-01080-t004:** Significant GO terms of DEGs on the basis of false discovery rate (FDR) corrected p-value of biological process in PI and GF stage in Ajay and Rajalaxmi hybrids.

S. No	GO Term	Annotation	Ajay		Rajalaxmi	
			PI	GF	PI	GF
			*p*-Value		*p*-Value	
1	GO:0009607	Response to biotic stimulus	1.9 × 10^−12^	0.0026	4.8 × 10^−6^	0.003
2	GO:0015979	Photosynthesis	1.1 × 10^−11^	0.008	NE *	4 × 10^−6^
3	GO:0006091	Generation of precursor metabolites and energy	2.6 × 10^−10^	0.0001	8.5 × 10^−5^	0.00019
4	GO:0006810	Transport	3.9 × 10^−10^	3.30 × 10^−10^	4.8 × 10^−5^	1.5 × 10^−7^
5	GO:0051234	Establishment of localization	3.9 × 10^−10^	3.30 × 10^−10^	4.8 × 10^−5^	1.5 × 10^−7^
6	GO:0051179	Localization	3.9 × 10^−10^	3.30 × 10^−10^	4.8 × 10^−5^	1.5 × 10^−7^
7	GO:0005975	Carbohydrate metabolic process	4.8 × 10^−10^	2.70 × 10^−14^	4.8 × 10^−10^	2.70 × 10^−14^
8	GO:0009628	Response to abiotic stimulus	6.3 × 10^−9^	2.20 × 10^−8^	NE *	1.1 × 10^−6^
9	GO:0050896	Response to stimulus	1.6 × 10^−6^	0.001	0.0032	0.0025
10	GO:0009987	Cellular process	2.8 × 10^−5^	0.015	NE *	0.0027

PI: Panicle Initiation; GF: Grain Filling; NE *: Not enriched.

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
