# Peer review of "Differential Expression of Genes at Panicle Initiation and Grain Filling Stages Implied in Heterosis of Rice Hybrids"

_ijms, 2020, doi:10.3390/ijms21031080_

Round 1

Reviewer 1 Report

I am aggreed this form of article.

Author Response

Agreed with publishing the current version.

Reviewer 2 Report

More descriptions on qRT-PCR, such as the number of genes, are needed in Methods and Results sections. Furthermore, the validation experiment and its corresponding results should be presented before other downstream analyses (e.g. mapping of the DEGs against those reported QTLs, TFs identification), right after the description on predicted DEGs. Avoid using strong definitive text when describing the findings. For instance, line 458 the statement “These results strongly suggest that the DEGs between hybrid and its parents most likely played significant contributory role in heterosis” needs to be rephrased, as any slight change in experimental conditions or environmental parameters would affect expression of the DEGs. This especially critical when the authors re-planted and used different input RNAs for RNAseq and qRT-PCR. Confusing statement needs to be clarified; e.g. “Both the hybrids and their parental lines were again sown during Rabi, 2019” (line 127).

Author Response

All corrections were made according to your instructions.

This manuscript is a resubmission of an earlier submission. The following is a list of the peer review reports and author responses from that submission.

Round 1

Reviewer 1 Report

I found the author had prove the better revision. I agreed with publishing the current version.

Reviewer 2 Report

1. The main issue of this study is still the absence of an independent validation (e.g. qRT-PCR) of at least some selected DEGs, without which would leave readers unsure about the quality, reproducibility and accuracy of the RNAseq analyses. With this, I’m not convinced that merely in silico analyses of the dataset would be reliable in all the downstream analyses, including the DEGs mapping against the yield-related QTLs, which should be a good connection with biological question posed in the study.

2. Given the authors preferred not to wait for the qRT-PCR work, which jeopardizes the quality of the work, I would recommend a rejection. 

3. Without the validation step, the second best options that the authors could do for improvement of the manuscript (for submission to perhaps a lower impact journal) could be: to include more descriptions of the RNAseq results, proving that these are of high quality (e.g. N50, Q20 or Q30), filtering parameters etc; include  schematic image to illustrate the DEGs locating within the QTLs intervals; and soften the wordings avoid using definitive text when describing the findings, e.g. lines 372-374 “These results strongly suggest that the DEGs between hybrid and its parents most likely played significant contributory role in heterosis”.